# Depression symptoms among University Students in the Khulna region of Bangladesh

**Habibur Rahman[1], Mortuja Mahamud Tohan** **[2]\*, Arifa Akter Easha[1], Nuzhat Fatema**[1,3‡]

**1** Development Studies Discipline, Khulna University, Khulna, Bangladesh, **2** School of General Education, Brac University, Dhaka, Bangladesh, **3** Department of Geography, Rutgers, The State University of New Jersey, New Jersey, United States of America

‡ This author is the senior author of this work.
\* mortuzacreations@gmail.com

## Abstract

University students frequently deal with different psychological changes in addition to coping with stressful academic and social obligations due to the facilities' environment, structure, and functions. This makes students more likely to experience depression, an eminent problem in today's world and particularly common among university studentsThe objective of this study was to evaluate the frequency of depression among university students in the Khulna region, taking into consideration the common symptoms of depression that occur with the transition from high school to university. The research involved a sample size of 1000 students representing from four universities located in the Khulna region. The random walk sampling technique was used for both sample selection and data collection. The Patient Health Questionnaire-9 was used to evaluate symptoms of depression. Descriptive statistics, chi-square followed by binary logistics regression were used to identify the associated factor contributing to the depression of the students. The findings show that 68% of students reported having moderate to severe depression, with female students reporting heightened rates (71.7%) than male students (62%). Binary logistic regression analysis results revealed that average and good result students reported depression difficulties that were 2.16 (95% CI 1.11-4.18) and 2.04 (95% CI 1.05-3.97) times greater than those of exceptional result students, respectively. The study finds an unusually high prevalence of depression among university students in the Khulna region. Further study should be conducted on the intricate factor such as anxiety, eating disorder, academic stress and social capital to better understand the overall factors responsible for this high prevalence of depression. Undoubtedly, there is a need to take immediate action to address and reduce the depression among university students through co-design approach.

## Introduction

Depression is recognized as a public health concern that presents with a variety of symptoms, including disruptions in sleep and eating patterns, diminished enjoyment in nearly all activities, neglect of self-care, impaired concentration, anxiety and disinterest, reduced energy levels, feelings of worthlessness and guilt, and difficulties in deductive thinking, focus, and decision-making in daily tasks [1,2]. University students are facing a growing prevalence of

**Data availability statement:** The datasets used and/or analysed during the current study uploaded as a supporting information file.

**Funding:** The authors received no specific funding for this work.

depression, which is a prevalent mental health issue that can range from mild melancholy to a severe and potentially life-threatening condition characterized by suicide ideation [3]. Currently, it is emerging as a significant public health issue due to several factors and is one of the primary contributors to poor health and disability. According to the World Health Organization (WHO), depression is listed as the fourth most significant contributor to disability globally and also reports that around 280 million individuals worldwide are affected by depression [4,5]. According to the Global Health Data Exchange, approximately 251-310 million individuals worldwide are affected by this issue. The World Health Organization has identified that this problem is more common in lower-and-middle income countries. In fact, they estimate that 76–85% of individuals with mental disorders in these countries do not have access to the required treatment [6–8]. Furthermore, the Mental Health Action Plan (2013–2030) established by the World Health Organization outlines a worldwide goal to promote and prevent mental health issues. On the other hand, Goals 3 and 4 of the Sustainable Development Agenda center on improving mental health and reducing mortality from non-communicable diseases via treatment and prevention [9]. Depression is prevalent among university students, who are a distinct demographic group experiencing a crucial transitional phase due to physical growth, more so than the general population [10]. However, Several research studies have indicated that university students globally often encounter despair, stress, psychological difficulties, and suicidal tendencies, among other challenges [1,11–14]. Bangladesh, a lower-middle income country in south Asia, has a prevalence of mental disorders ranging from 6.5% to 31.0% for adults (18–65 years) and 13.4% to 22.9% for children (5–17 years). This problem is becoming more and more important for public health since it not only affects development in this nation but also raises suicide and other concomitant psychiatric disease risks [15,16]. A significant percentage of students attending universities in Bangladesh are currently facing mental health issues. Previous research has identified several factors that may contribute to students' susceptibility to depression, including difficulties in adapting to a new environment, academic pressure, dissatisfaction with academic performance, challenges with accommodation, experiences of ragging or bullying, concerns about career prospects, high expectations from parental figures, interpersonal difficulties, and family issues [10,17–19]. Furthermore, lifestyle issues such as the absence of traditional adult supervision, lack of conventional social assistance, financial challenges, smoking, substance abuse, and physical and mental distress [20–24]. Prior studies conducted among learners in universities have shown significant levels of distress, For example, a study conducted in Bangladesh found that 52% of the participants reported experiencing depression, which was the most prevalent psychiatric morbidity. Additionally, other psychiatric morbidities such as personality disorder (14.29%) and anxiety disorder (8.93%) were also reported [25]. A further survey conducted among undergraduate applicants in Bangladesh revealed that a significant proportion of students encountered stress levels that comprised mild to extremely extreme signs of depression, anxiety, and stress, with percentages of 57.7%, 61.4%, and 44.6%, consecutively [15]. Furthermore, a study conducted on medical students revealed that the prevalence of depression varied across severity levels, with rates of 3.6%, 14.5%, and 20.8% for severe, moderate, and mild depression, each. Simultaneously, another study focusing on first-year university students found that the prevalence rates of moderate to extremely severe levels of depression and anxiety were 69.5% and 61%, correspondingly [26,27]. A number of earlier research conducted in Bangladesh also indicated that university students' low socioeconomic level, gender, academic achievement, high unemployment rate, and uncertain future all play a part in the onset of depression symptoms [27–29]. Bangladesh's government created a national strategy for adolescent health to address issues between 2017 and 2030, stressing that lack of knowledge makes it difficult to create suitable treatments to raise teenagers' mental health [30]. Hence, it is imperative to examine

depressive symptoms in adolescents as promptly recognizing such symptoms can effectively prevent the development of the condition and enable the formulation of solutions to mitigate the symptoms that may arise during the ongoing crisis [31,32]. During the COVID-19 pandemic in Khulna, two investigations were carried out. One study indicated that 32.6% of the participants suffered from mental depression, while 44.9% reported feeling anxious. The other study indicated that the majority of the participants experienced moderate to severe levels of anxiety, depression, and mental stress. Depression was also substantially associated with age, gender, residence, falling behind academically, contracting COVID-19, losing tuition, and being unsure of one's professional path [33,34]. Based on the literature reviewed, it is evident that there have been limited studies conducted on adolescent depressive symptoms in Khulna, Bangladesh. To the best of the authors' knowledge, there is a need for further research studies to investigate the prevalence and factors associated with emotional disorders among university students. This study's overarching objective was to investigate the extent to which depression is widely accepted among university students in Khulna. The study also sought to investigate the primary risk factors associated with depression and the connection between depression and gender, employment status, married and divorced/separated students, co-curricular activities, and regular religious activities among the students.

## Methodology

### Research method

A quantitative approach was taken in this investigation. This study employed a cross-sectional survey, a research approach that collected data from a specific population at a single point in time. The purpose of the study was to assess the prevalence of depression among university students in Khulna. During the months of June, July, and August of 2023, researchers conducted surveys with students enrolled in four universities that were specifically chosen from the Khulna area. The rationale for purposive selection is to deliberately choose a particular group of individuals or units for analysis. This is done based on the assumption that, considering the goals and objectives of the study, certain types of people may possess distinct and significant perspectives on the ideas and issues being investigated. Consequently, it is necessary to include them in the sample [35].

**Study area.** Khulna District (see Fig 1) Latitude and longitude coordinates are: Latitude, 22° 30"N; Longitude, 89° 30"E. Khulna University Latitude, 22° 48' 05"N; Longitude, 89° 31' 55"E, Khulna University of Engineering & Technology Latitude, 22° 54' 05"N; Longitude, 89° 30' 05"E. Khulna Agricultural University Latitude, 22° 50' 55"N; Longitude, 89° 30' 15"E. Kushtia District latitude and longitude, Latitude, 24° 89"N; Longitude, 88° 45'E. Islamic University latitude and longitude, Latitude, 23° 43' 20"N; Longitude, 89° 09' 05""E. Depression is the foremost mental health issue confronting students, adversely affecting their academic performance and overall well-being, which in turn jeopardizes their relationships with peers and family. For instance, a significant proportion of students at Khulna University experience moderate to severe levels of depression, anxiety, and mental stress, resulting in financial losses related to tuition and uncertainty regarding career pursuits, while simultaneously lagging academically. Students at the Islamic university exhibited moderate levels of despair [34,36].

### Study population

The study focused on university students enrolled at the university. The study included both male and female students from various institutions who were attending the university during the study period.

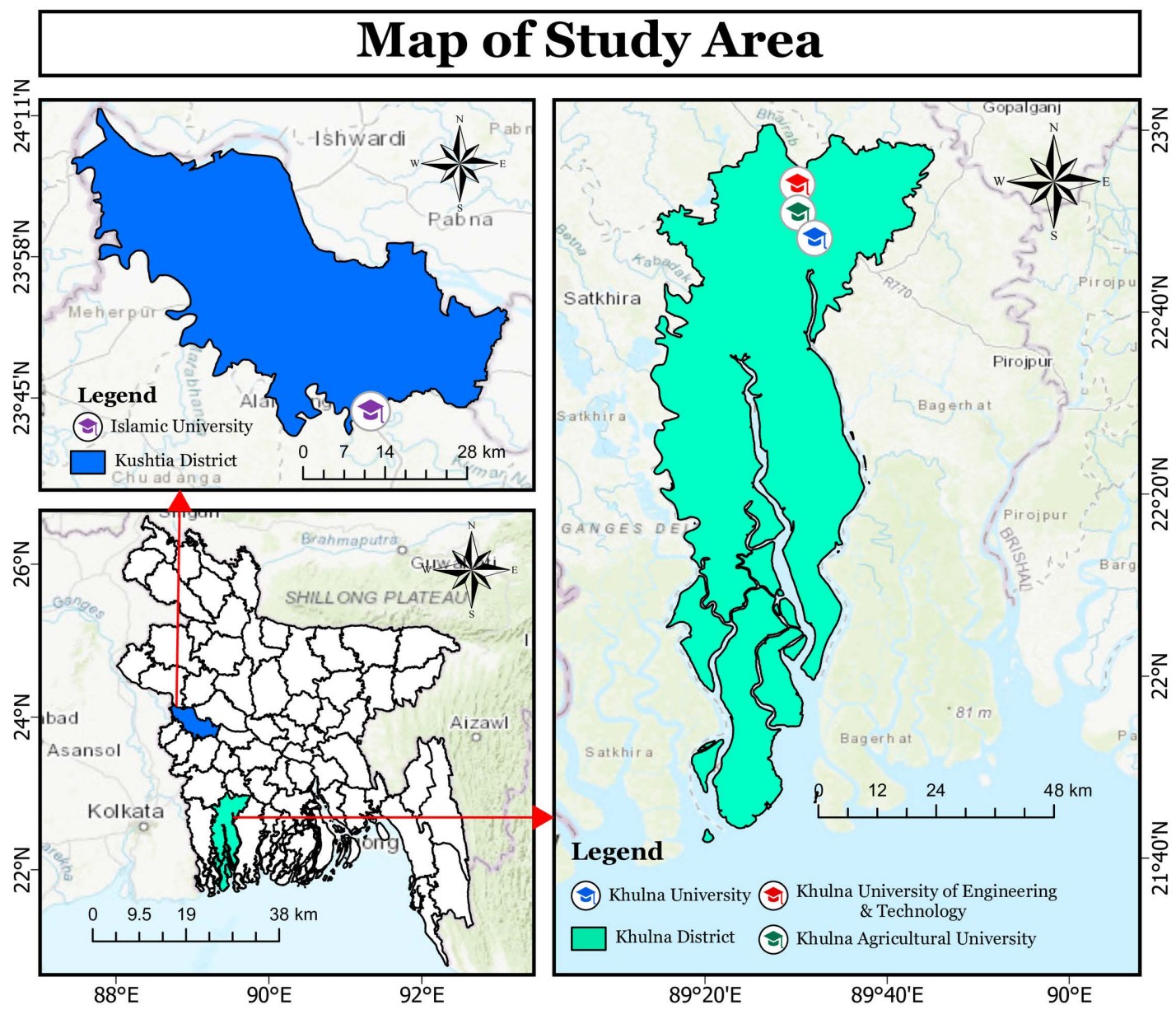

**Fig 1. Map of study area (Khulna district).**

### Inclusion and exclusion criteria

The criteria for inclusion in the study were obtaining agreement to participate and being a current university student from any field of study. Furthermore, there were no limitations regarding the study's year or field. Students with a history of mental illness diagnoses or who had previously undergone any screening tests were not included in the study.

### Sampling and sampling procedure

Four universities in Khulna were selected randomly. The ultimate analysis incorporated data from the entire cohort of 1000 (see Table 1) students who provided responses to the

**Table 1. Sampling procedure.**

| University name | Total students | Sampling |
|---|---|---|
| Khulna University | 7,644 | 520 |
| Khulna University of Engineering & Technology | 5,240 | 190 |
| Khulna Agricultural University | 504 | 30 |
| Islamic University | 16,000 | 260 |

**Source:** www.ku.ac.bd, https://www.kuet.ac.bd/, https://kau.ac.bd/, https://www.iu.ac.bd/.

questionnaire within the specified data collection timeframe. The sample consisted of 597 males (59.7%) and 403 females (40.3%). Here stratified random sampling technique was adopted. It selects specific kinds or groups of participants that need to be part of the final sample. The sample is then stratified by the characteristic of the participant or group, with a specific number allocated to each stratification[35].

The sample size of the study was determined by using the formula of Godden. According to apply formula of [37]. The Sample size (s) was calculated as follows:

$$s = \frac{z^2 p(1-p)}{M^2}$$

$$s = \frac{1.96^2 \times 0.03519(1-0.3519)}{(0.01142)^2}$$

$$s = \frac{0.13042}{(0.01142)^2}$$

$$s = 1000.09$$

$$\approx 1000$$

Where,

S = Sample size.

Z = Standard normal Deviation set 95% confidence level = 1.96

P = Percentage of Population picking a choice, expressed as decimal = 3.519% = 0.3519

M = Margin of error 1.1142% = 0.1142

## Data collection

To gather information from the participants, a self-designed and pilot-tested partially structured questionnaire was utilized. Before commencing data collection, the data collection team underwent comprehensive training lasting one full day. The training was conducted in the presence of a lead supervisor and other experts from Khulna University. The main focus of the training included ethical considerations and the implementation of a consent-based survey technique. Additionally, a pilot study involving 50 students was conducted to refine the questionnaire before its finalization.

## Measures

The PHQ-9 is a 9-item self-report questionnaire that assesses depression symptoms experienced by participants during the past 2 weeks [21]. Furthermore, it evaluates not only severe

depression but also the beginning of depressive disorder in epidemiological research including the general population. The PHQ-9 has been validated in multiple studies conducted in primary care; medical outpatients; and specialist medical services in different cultures. It has been found to be acceptable and is as good as longer clinician-administered instruments in a range of settings, countries, and populations. Each question was answered using a four-point Likert scale where respondents could choose from zero (Never) to three (Almost daily) options. It includes nine questions that gauge the frequency and seriousness of symptoms like losing interest in activities, feeling depressed or hopeless, and having trouble falling asleep or focusing. Participants had to describe their feelings from the previous two weeks on the PHQ-9 questionnaire. Higher total scores indicated more severe depression, ranging from 0 to 27, where score ranging 1-4 indicates Minimal depression, 5-9 indicates Mild depression 10-14 indicates Moderate depression, 15-19 indicates moderately severe depression and 20-27 indicates severe depression. However, a score of 10 or higher indicated that person has depression and lower or equal to 10 signifies person without depression. The dependent variable, depression, in our study, was categorized as either having depression (total score > 10) or not (total score ≤10) [38].

**Statistical analysis.** The aim of this study was to assess the prevalence of depression among university students and the primary risk factors associated with depression. Descriptive statistics were employed to elucidate the data and discern its inherent patterns. Bivariate analysis was conducted to assess the nature of the relationship between independent factors and the dependent variables. The empirical link between perceptions of the dependent variable, depression, and other covariates was determined using Chi-Square tests. Subsequently, binary logistic regression analysis was performed to predict the relationship and likelihood of groups of independent variables influencing the presence of depression. SPSS version 26 and Microsoft excel were utilized to carry out the analysis.

People of various genders experience cultural variances in social stigma, gender differences, personality traits, and educational environments differently, which contributes to gender-based mental health differences[39,40]

This result shows disagreement with findings from several studies that claim there was no significant difference among participants in their depression symptoms, anxiety, and stress levels regarding their marital status [15,41].

Depression is more common among graduate students than undergraduates, according to this study. This is due to factors such as high parental expectations, family conflicts, a lack of financial support, worries about the future, and a toxic psychological environment. Too often, parents worry about what other people think of them, stress over their inability to secure gainful employment, and supervise their children's life [42,43].

## Results

Table 2 Our survey of 1,000 participants indicated that 68% of university learners suffer from depression.

There was no significant association between age (22 or younger vs. older than 22) and depression symptoms ($\chi2 = 1.671$, $p = 0.196$). This shows that age alone may not be a reliable indicator of depression in this study. Females expressed higher rates of depression compared to males ($\chi2 = 10.148$, $p < 0.001$). Marital status was found to be a significant predictor of depression symptoms, with married and divorced/separated individuals showing higher rates than unmarried individuals ($\chi2 = 5.998$, $p = 0.030$). Participants with unemployed fathers were significantly more likely to express depression than those with employed fathers ($\chi2 = 9.632$, $p = 0.002$). Participants with employed mothers had significantly higher levels of depression symptoms ($\chi2 = 26.020$, $p<0.001$).

**Table 2.  Frequency and association of factor categories with depression symptoms.**

| Variables | Total (%) | Status of Depression | | Chi-square value | P-value |
|---|---|---|---|---|---|
| | | No | Yes | | |
| **Age group** | | | | 1.671 | 0.196 |
| 22 or less | 520 (52.0) | 187 (36.0) | 333 (64.0) | | |
| 22+ | 480 (48.0) | 154 (32.1) | 326 (67.9) | | |
| **Sex** | | | | 10.148 | < 0.001 |
| Male | 597 (59.7) | 227 (38.0) | 370 (62.0) | | |
| Female | 403 (40.3) | 114 (28.3) | 289 (71.7) | | |
| **Marital Status** | | | | 5.998 | 0.030 |
| Unmarried | 868 (86.8) | 308 (35.5) | 560 (64.5) | | |
| Married | 111 (11.1) | 29 (26.1) | 82 (73.9) | | |
| Divorced/separate | 21 (2.1) | 4 (19.0) | 17 (81.0) | | |
| **Father occupation** | | | | 9.632 | 0.002 |
| Employed | 839 (83.9) | 269 (32.1) | 570 (67.9) | | |
| Not Working | 161 (16.1) | 72 (44.7) | 89 (55.3) | | |
| **Mother Occupation** | | | | 26.020 | < 0.001 |
| Housewife | 711 (71.1) | 277 (39.0) | 434 (61.0) | | |
| Employed | 187 (18.7) | 43 (23.0) | 144 (77.0) | | |
| Not Working | 102 (10.2) | 21 (20.6) | 81 (79.4) | | |
| **Came from** | | | | 2.554 | 0.002 |
| Rural | 599 (59.9) | 216 (36.1) | 383 (63.9) | | |
| Urban | 401(40.1) | 125 (31.2) | 276 (68.8) | | |
| **Place of residence** | | | | 5.201 | 0.004 |
| Home | 191(19.1) | 67 (35.1) | 124 (64.9) | | |
| Mass | 282 (28.2) | 81 (28.7) | 201 (71.3) | | |
| Student hall | 527(52.7) | 193 (36.6) | 334 (63.4) | | |
| **Studied in major** | | | | 3.590 | 0.084 |
| Science | 425(42.5) | 153 (36.0) | 272 (64.0) | | |
| Commerce | 99(9.9) | 26 (26.3) | 73 (73.7) | | |
| Humanities | 476(47.6) | 162 (34.0) | 314 (66.0) | | |
| **Education level of participant/Last achieved** | | | | 2.644 | < 0.001 |
| Undergraduate | 897(89.7) | 313 (34.9) | 584 (65.1) | | |
| Postgraduate | 103(10.3) | 28 (27.2) | 75 (72.8) | | |
| **Academic performance of CGPA** | | | | 3.887 | < 0.001 |
| Average | 519(51.9) | 175 (33.7) | 344 (66.3) | | |
| Good | 429(42.9) | 142 (33.1) | 287 (66.9) | | |
| Excellent | 52 (5.2) | 28 (53.8) | 24 (46.2) | | |
| **Engaged with co-curriculum activities** | | | | 8.978 | 0.003 |
| Yes | 468 (46.8) | 286 (61.1) | 182 (38.9) | | |
| No | 532 (53.2) | 159 (29.9) | 373 (70.1) | | |
| **Regularly do any sports\physical exercise** | | | | 4.639 | < 0.001 |
| Yes | 387 (38.7) | 240 (62.0) | 147 (38.0) | | |
| No | 613 (61.3) | 194 (31.6) | 419 (68.4) | | |
| **Playing cards at night for long** | | | | 5.573 | 0.002 |
| Yes | 249 (24.9) | 80 (32.1) | 169 (67.9) | | |
| No | 751 (75.1) | 490 (65.2) | 261 (34.8) | | |
| **Involved in regular religious activities** | | | | 6.598 | < 0.001 |
| Yes | 627 (62.7) | 395 (63.0) | 232 (37.0) | | |

*(Continued)*

**Table 2.** (Continued)

| Variables | Total (%) | Status of Depression | | Chi-square value | P-value |
|---|---|---|---|---|---|
| | | **No** | **Yes** | | |
| No | 373 (37.3) | 109 (29.2) | 264 (70.8) | | |
| **Use social media** | | | | 0.159 | 0.190 |
| Yes | 925 (92.5) | 317 (34.3) | 608 (65.7) | | |
| No | 75 (7.5) | 24 (32.0) | 51 (68.0) | | |
| **Smoke** | | | | 3.740 | < 0.001 |
| Yes | 239 (23.9) | 76 (31.8) | 163 (68.2) | | |
| No | 761 (76.1) | 496 (65.2) | 265 (34.8) | | |
| **Relationship Status** | | | | 1.435 | 0.231 |
| Single | 797 (79.7) | 279 (35.0) | 518 (65.0) | | |
| Engaged | 203 (20.3) | 62 (30.5) | 141 (69.5) | | |
| **Worried about the future** | | | | 0.581 | 0.003 |
| Yes | 542 (54.2) | 190 (35.1) | 352 (64.9) | | |
| No | 458 (45.8) | 307 (67.0) | 151 (33.0) | | |
| **Difficulty sleeping for roommates** | | | | 4.879 | <0.001 |
| Yes | 716 (71.6) | 240 (33.6) | 475 (66.4) | | |
| No | 284 (28.4) | 183 (64.4) | 101 (35.6) | | |
| **Take a light nap in the afternoon** | | | | 0.001 | 0.973 |
| Yes | 555 (55.5) | 189 (34.1) | 366 (65.9) | | |
| No | 445 (44.5) | 152 (34.2) | 293 (65.8) | | |
| **Sleep at night under the stress of thesis/project/exam/class assessment** | | | | 0.011 | 0.915 |
| Yes | 593 (59.3) | 203 (34.2) | 390 (65.8) | | |
| No | 407 (40.7) | 138 (33.9) | 269 (66.1) | | |
| **Income** | | | | 3.489 | 0.002 |
| <=20000 | 289 (28.9) | 95 (32.9) | 194 (67.1) | | |
| 20000<income=<40000 | 467 (46.7) | 172 (36.8) | 295 (63.2) | | |
| 40000+ | 244 (24.4) | 170 (69.7) | 74 (30.3) | | |
| **Expenditure** | | | | 1.503 | 0.050 |
| <=20000 | 333 (33.3) | 120 (36.0) | 213 (64.0) | | |
| 20000<expenditure<=40000 | 466 (46.6) | 159 (34.1) | 307 (65.9) | | |
| 40000+ | 201 (20.1) | 139 (69.2) | 62 (30.8) | | |
| **Earning hand** | | | | 1.884 | 0.003 |
| <1 | 530 (53.)0 | 191 (36.0) | 339 (64.0) | | |
| 2+ | 470 (47.0) | 320 (68.1) | 150 (31.9) | | |
| **Use Facebook time** | | | | 3.771 | 0.152 |
| <1 | 114 (11.4) | 43 (37.7) | 71 (62.3) | | |
| 2-3 | 390 (39.0) | 119 (30.5) | 271 (69.5) | | |
| 4+ | 496 (49.6) | 179 (36.1) | 317 (63.9) | | |

Urban residents showed higher rates of depression symptoms compared to rural residents ($\chi 2 = 2.554$, p = 0.002). The type of residence (home, mass housing, or student hall) was significantly associated with depression symptoms, with those in mass housing having the highest rates ($\chi 2 = 5.201$, p = 0.004). There was no significant association observed between academic major (science, commerce, humanities) and depression ($\chi 2 = 3.590$, p = 0.084). Depression symptoms differed significantly between education levels, with postgraduate students having higher rates than undergraduates ($\chi 2 = 2.644$, p<0.001).

Academic performance was significantly associated with depression, with participants disclosing high scores being less likely to experience depression than those with average or good grades (χ2 = 3.887, p<0.001). Participants who participated in co-curricular activities had significantly lower levels of depression symptoms compared to those who did not (χ2 = 8.978, p = 0.003). Regular physical activity was significantly associated with lower rates of depression, χ2 = 4.639, p<0.001. Participants who played cards at night had significantly higher depression symptoms (χ2 = 5.573, p = 0.002).

Regular participation in religious activities was linked to lower depression symptoms (χ2 = 6.598, p<0.001). There was no significant association between social media use and depression symptoms (χ2 = 0.159, p = 0.190). Smoking was significantly associated with increased depression symptoms (χ2 = 3.740, p<0.001). There was no significant correlation between relationship status (single or engaged) and depression symptoms (χ2 = 1.435, p = 0.231).

Participants who were concerned about the future showed significantly higher depression symptoms (χ2 = 0.581, p = 0.003).

Roommates' difficulty sleeping was significantly associated with higher depression symptoms (χ2 = 4.879, p<0.001. There was no significant correlation between taking a light afternoon nap and depression symptoms (χ2 = 0.001, p = 0.973). There was no significant correlation between academic stress-induced sleep problems and depression symptoms (χ2 = 0.011, p = 0.915). Lower-income participants reported significantly more depression symptoms than those in higher income brackets (χ2 = 3.489, p = 0.002).

Reduced spending was linked to higher levels of depression symptoms (χ2 = 1.503, p = 0.050). Individuals in households with two or more earners reported fewer depression symptoms (χ2 = 1.884, p = 0.003). Time spent on Facebook did not significantly correlate with depression symptoms (χ2 = 3.771, p = 0.152).

Table 3 exhibits the binary logistic regression analysis with adjusted odds and unadjusted odds ratios at a 95% confidence interval of the participant's depression status. Males had significantly lower odds of experiencing depression symptoms than females in both adjusted (OR = 0.56, 95% CI [0.41, 0.79], p < 0.001) and unadjusted models (OR = 0.64, 95% CI [0.49, 0.84], p < 0.001). Participants with employed fathers had higher odds of depression symptoms than those with unemployed fathers in both adjusted (OR = 1.99, 95% CI [1.36, 2.92], p < 0.001) and unadjusted models (OR = 1.71, 95% CI [1.22, 2.42], p = 0.002). Rural participants were less likely to report depression than urban participants, with significant differences in adjusted (OR = 0.86, 95% CI [0.62, 1.18], p = 0.004) and unadjusted models (OR = 0.80, 95% CI [0.61, 1.05], p = 0.010).

There were no significant differences in depression symptoms between education levels in either adjusted (OR = 1.02, 95% CI [0.59, 1.76], p = 0.954) or unadjusted models (OR = 0.70, 95% CI [0.44, 1.10], p = 0.120). Participants with average or good grades were more likely to suffer from symptoms of depression than those with excellent grades. For average grades, adjusted OR = 2.16 (95% CI [1.11, 4.18], p = 0.003), while unadjusted OR = 1.69 (95% CI [0.95, 2.10], p < 0.001). For good grades, adjusted OR = 2.04 (95% CI [1.05, 3.97], p = 0.001); unadjusted OR = 1.73 (95% CI [0.10, 3.01], p = 0.005).

Co-curricular activity participants had lower odds of depression symptoms, with significant associations in both adjusted (OR = 0.74, 95% CI [0.55, 1.01], p < 0.001) and unadjusted models (OR = 0.67, 95% CI [0.52, 0.87], p = 0.003). In the unadjusted model, regular physical exercise was associated with lower odds of depression (OR = 0.76, 95% CI [0.58, 0.99], p = 0.040), but after adjustment, the relationship was marginally significant. Participation in certain activities, such as playing cards at night, was associated with higher odds of depression symptoms in both adjusted (OR = 1.01, 95% CI [0.70, 1.47], p = 0.006) and unadjusted models (OR = 1.13, 95% CI [0.83, 1.57], p = 0.009). Regular religious activities were linked

**Table 3.** Logistic regression analysis of risk factors for depression symptoms yields odds ratios and 95% confidence intervals.

| Variable (Depression) | Adjusted (95%CI) | Sig | Unadjusted(95%CI) | sig |
|---|---|---|---|---|
| **Sex** | | | | |
| female | 1 | | 1 | |
| Male | 0.56 (0.41–0.79) | <0.001 | 0.64 (0.49–0.84) | < 0.001 |
| **Father occupation** | | | | |
| Not Working | 1 | | 1 | |
| Employed | 1.99 (1.36–2.92) | <0.001 | 1.71 (1.22–2.42) | 0.002 |
| **Came from** | | | | |
| Urban | 1 | | 1 | |
| Rural | 0.86 (0.62–1.18) | 0.004 | 0.80 (0.61–1.05) | 0.010 |
| **Education level of participant/Last achieved** | | | | |
| Postgraduate | 1 | | 1 | |
| Undergraduate | 1.02 (0.59–1.76) | 0.954 | 0.70 (0.44–1.10) | 0.120 |
| **Academic performance of CGPA** | | | | |
| Excellent | 1 | | 1 | |
| Average | 2.16 (1.11–4.18) | 0.003 | 1.69 (0.95 -2.10) | < 0.001 |
| Good | 2.04 (1.05–3.97) | 0.001 | 1.73 (0.10–3.01) | 0.005 |
| **Engaged with co-curriculum activities** | | | | |
| No | 1 | | 1 | |
| Yes | 0.74 (0.55–1.01) | <0.001 | 0.67 (0.52–0.87) | 0.003 |
| **Regularly do any sports\physical exercise** | | | | |
| No | 1 | | 1 | |
| Yes | 0.74 (0.54 - 1.02) | 0.066 | 0.76 (0.58–0.99) | 0.040 |
| **Playing cards at night for long** | | | | |
| No | 1 | | 1 | |
| Yes | 1.01 (0.70–1.47) | 0.006 | 1.13 (0.83–1.57) | 0.009 |
| **Involved in regular religious activities** | | | | |
| No | 1 | | 1 | |
| Yes | 0.68 (0.50–0.92) | 0.003 | 0.70 (0.53–0.93) | 0.002 |
| **Use social media** | | | | |
| No | 1 | | 1 | |
| Yes | 0.97 (0.53–1.78) | 0.920 | 0.90 (0.55–1.49) | 0.690 |
| **Smoke** | | | | |
| No | 1 | | 1 | |
| Yes | 1.21 (0.84 -1.76) | 0.002 | 1.15 (0.84–1.56) | < 0.001 |
| **Relationship Status** | | | | |
| No | 1 | | 1 | |
| Yes | 0.86 (0.59 -1.27) | 0.451 | 0.82 (0.59–1.14) | 0.231 |
| **Worried about the future** | | | | |
| No | 1 | | 1 | |
| Yes | 1.05 (0.78–1.41) | 0.767 | 0.91 (0.70–1.19) | 0.488 |
| **Difficulty sleeping for roommates** | | | | |
| No | 1 | | 1 | |
| Yes | 0.96 (0.69–1.34) | <0.001 | 0.92 (0.69–1.05) | 0.003 |
| **Sleep at night under the stress of thesis/project/exam/class assessment** | | | | |
| No | 1 | | 1 | |
| Yes | 1.10 (0.81–1.48) | 0.546 | 0.99 (0.76–1.27) | 0.915 |
| **Income** | | | | |

*(Continued)*

**Table 3.** (Continued)

| Variable (Depression) | Adjusted (95%CI) | Sig | Unadjusted(95%CI) | sig |
|---|---|---|---|---|
| 40000+ | 1 | | 1 | |
| <=20000 | 2.03 (0.79–5.25) | 0.143 | 0.89 (0.62–1.28) | 0.530 |
| 20000<income=<40000 | 0.89 (0.44–1.78) | 0.736 | 0.75 (0.54–1.04) | 0.084 |

to significantly lower odds of depression, both adjusted (OR = 0.68, 95% CI [0.50, 0.92], p = 0.003) and unadjusted (OR = 0.70, 95% CI [0.53, 0.93], p = 0.002). No significant relationship was identified between social media use and depression.

Smoking was linked to a higher risk of depression in both adjusted (OR = 1.21, 95% CI [0.84, 1.76], p = 0.002) and unadjusted (OR = 1.15, 95% CI [0.84, 1.56], p < 0.001) models. Relationship status, future worries, and academic-related sleep disturbances had no significant associations with depression. In the adjusted model, participants with lower incomes (≤20000) had higher odds of depression, but this was not statistically significant (OR = 2.03, 95% CI [0.79, 5.25], p = 0.143).

## Discussion

The primary focus of this study was to evaluate Depression and its main risk factors, including socio-demographic factors, financial challenges, smoking, and marital status, among university students of Khulna. It builds upon previous research by offering additional evidence to confirm the high occurrence of depression among students in the context of Bangladesh [44–46]. The findings of our study indicate that among the 1000 participants, a significant proportion (68%) suffered from depression. Furthermore, our sample of students in Bangladesh has consistently exhibited elevated levels of depressive symptoms [47–51]. The results showed that there is a statistically significant relationship between gender and depression, and our study also demonstrated that both males and females experience different levels of depression [52,53]. Nevertheless, there are research that contradict our findings and have not discovered any gender differences in depression [46,54–56]. Consistent with the results of other studies among students at universities, our data show that female students have a higher prevalence of depression compared to male students [57–59]. People of various genders experience cultural variances in social stigma, gender differences, personality traits, and educational environments differently, which contributes to gender-based mental health differences. Our research revealed that married students exhibited a higher vulnerability to depression in comparison to their single counterparts. This could be attributed to the fact that married students encounter a greater number of demanding circumstances compared to their single counterparts. These may include challenges related to managing their time effectively, handling household responsibilities, raising children, and seeking permission from their spouse for various matters. This finding is consistent with prior research that has highlighted the higher levels of depression among married and divorced students [60]. Furthermore, the primary risk factors associated with depression in married women include having many marriages, experiencing a poor relationship with one's husband, engaging in hard work, and having chronic medical comorbidity [61]. This result shows disagreement with findings from several studies that claim there was no significant difference among participants in their depression symptoms, anxiety, and stress levels regarding their marital status. Contrary to our study, [54] highlighted that Depression is a real problem among students who are not living with a partner. Depression is more common among graduate students than undergraduates, according to this study. This is due to factors such as high

parental expectations, family conflicts, a lack of financial support, worries about the future, and a toxic psychological environment. Too often, parents worry about what other people think of them, stress over their inability to secure gainful employment, and supervise their children's life. The primary factors contributing to this disparity include apprehension regarding future prospects, employment opportunities, familial expectations, and societal obstacles [55]. Additionally, [62] emphasized that depression may arise as a result of job preparation, workload, and limited leisure time, particularly after completing one's education. Contrary to earlier research, a study conducted among Turkish university students found that younger students had higher levels of stress compared to their senior counterparts [60]. Studies conducted in Iran, Jordan, and Bangladesh have discovered that students who enroll at university and leave their families for the first-time experience various changes, including a new environment, hazing, separation from family, changes in friendships, and academic pressure. These changes can potentially contribute to the development of depression [10,54,63]. Participants who exercised at least once a day had much lower depression levels than those who did not exercise regularly, and the study found a strong correlation between physical activity and depression. This finding is in line with earlier studies that highlighted the importance of physical activity in preventing the onset of depression symptoms in teenagers [64,65]. However, some study came to the opposite conclusion, finding no evidence that exercise prevented students from experiencing depressive symptoms [47,66]. The study found that most of the families of the respondents had moderate to low income. This financial situation could cause students to be concerned about their academic performance, the cost of education, economic difficulties, and feelings of insecurity. Ultimately, these factors may contribute to the high occurrence of depression in the study population. Contrary to this, another study indicated that adolescents from higher-income families were less concerned about their academic performance and more likely to engage in substance abuse and dangerous behaviors [45,67]. The current study revealed that being involved in a relationship was identified as a predictor for experiencing elevated levels of depression. This association may be attributed to the fact that individuals in relationships allocate a significant amount of time to their partners, which can negatively impact their academic performance and contribute to an increase in depressive symptoms. The outcome was consistent with the findings of the prior investigation [50]. Furthermore, our research has revealed a correlation between smoking and depression, as smoking serves as a temporary coping technique for stress [50,68]. The analysis of this study indicated that academic achievement played a significant role, as individuals with average and good results displayed a stronger correlation with the issue of depression compared to students who achieved excellent grades. The research undertaken in Ethiopia and Pakistan consistently reported the same finding [69,70]. Research has demonstrated that students who frequently participated in religious activities had a reduced likelihood of developing depression compared to those who did not engage in such activities. In summary, our data revealed a correlation between engagement in religious activities and the presence of depression. The findings of our research corroborate the conclusions drawn in other investigations. As an illustration, [71] observed Several articles have revealed a strong association between religious practices and the amelioration of depressed symptoms. A separate study emphasized that there was a substantial and positive correlation between religiosity and depression. Moreover, they employed their religious beliefs and practices as a means to alleviate their feelings of depression [72]. Nevertheless this goes against what was found in a study by [73] who found no correlation between depressive symptoms and religious affiliation or activity frequency.

However, the investigation was not without its limitations. Initially, data was gathered from a comparatively limited subset of students. The lack of mandatory participation precludes

the ability to extrapolate these findings to alternative contexts or populations. Additionally, as previously stated, the design of this cross-sectional study may pose challenges in precisely establishing a correlation between the variables under investigation and depression. Therefore, further investigation is necessary to collect data from various sources in order to encompass a wide range of issues and experiences in a comprehensive, nationwide study.

One disadvantage of the study is its cross-sectional study design, which hinders the capacity to establish cause-and-effect correlations and limits causal inference. Nevertheless, in order to mitigate this constraint, we supplemented our discoveries with preexisting literature to offer a more comprehensive framework and propose potential avenues for future longitudinal studies. This study used self-reported data, which could have been susceptible to over-reporting. To mitigate this, participants were encouraged to provide sincere and thoughtful responses. Additionally, statistical techniques were employed to address potential biases. Furthermore, comprehensive data checks were conducted to ensure the reliability of the findings. Recognizing these constraints, it can be contended that additional research is necessary to thoroughly investigate the factors linked to the mental health challenges experienced by young students. Simultaneously, comprehensive methodologies should be employed to improve the accuracy and relevance of the findings. The findings of our study can be utilized for conducting more extensive research, improving the prevailing mental health crises, and making well-informed decisions, which can be regarded as a significant achievement of this research.

## Conclusion

The current investigation unveiled a substantial proportion of university students in Bangladesh experiencing signs of depression. The study revealed that experiencing negative mental health symptoms was more common among female students, those from lower-income families, those who preferred public universities, those who had marital pressure, those who had experienced stressful life events in the past, and those who had attempted suicide in the past. The results of this study revealed substantial associations between depression and stress with gender, relationship status, participation in extracurricular activities, engagement in physical activities, involvement in religious activities, and time spent online. Therefore, it is crucial for students to priorities these influential factors and take appropriate action to overcome and avoid depression and stress. This can be achieved by actively engaging in activities that have been proven effective in reducing depression and stress, such as participating in extracurricular activities and engaging in physical and religious practices. Furthermore, individuals should ensure they are well-informed about the current circumstances and refrain from panicking over unverified reports. Besides, the appropriate Bangladeshi authorities (including academic administrations, policymakers, government organizations, and health experts) need to address these issues to create a setting that is friendly to students as well as to design effective mental health intervention programs that guard against depression issues and also assist students in increasing their self-efficacy. Since this was an online survey, a clinical examination to confirm the existence of depression was not available. Therefore, we suggested that future researchers use clinical assessment to verify the occurrence of depression. Our findings further highlight the importance of conducting a large longitudinal study to investigate the causes of depression among university students in Bangladesh. Additionally, it is necessary to conduct validation studies in order to create and authenticate dependable tools for implementation among university students in Bangladesh. The findings from this study, such as the factors that contribute to depression, may have practical applications for clinical psychiatric nursing practices that are currently addressing mental health issues among university students.

## Strengths and limitations

An advantage of the current study is that we gathered data from four institutions, which broadens the scope of the sample and offers diverse viewpoints. Another advantage was our ability to gather internet data, enabling us to quickly reach students.

Several limitations of the current investigation must be acknowledged. Firstly, as a cross-sectional study, it collected data at a singular point in time, limiting its ability to establish cause-and-effect relationships. Secondly, the study relied on an online survey approach, resulting in potential sampling biases because it included only respondents with internet access. Therefore, the cohort may not be representative of Bangladesh's total population. Thirdly, the study lacked an assessment of depression symptoms quality, which could have yielded more comprehensive results. With a standardized diagnostic threshold for depression quality, it may be easier to assess depression symptoms accurately. Fourthly, information was collected from participants at four different institutions, making it difficult to generalize the findings to the general population. In addition, self-reporting, the method used in the study, has inherent limitations, such as social desirability and memory recall biases. These limitations should be considered when interpreting the study's results and deriving conclusions.

## Supporting information

**S1 Data. SPSS dataset.**
(SAV)

## Acknowledgements

I appreciate your help the authors who wrote this study want to thank all the people who took the time to fill out the investigation and give their important information. Without their help, this study would not have been possible. The authors also want to thank Assistant Professor Md. Ashfikur Rahman for meticulously verifying any discrepancies in this work.

## Author contributions

**Conceptualization:** Habibur Rahman.

**Data curation:** Habibur Rahman, Mortuja Mahamud tohan.

**Formal analysis:** Habibur Rahman, Mortuja Mahamud tohan, Arifa Akter Easha.

**Methodology:** Habibur Rahman, Mortuja Mahamud tohan, Nuzhat Fatema.

**Resources:** Habibur Rahman, Mortuja Mahamud tohan.

**Supervision:** Nuzhat Fatema.

**Validation:** Arifa Akter Easha.

**Writing – original draft:** Habibur Rahman, Mortuja Mahamud tohan, Arifa Akter Easha.

**Writing – review & editing:** Nuzhat Fatema.

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
