## [Decision Letter · Decision Letter 0]

3 Jul 2024

PMEN-D-24-00166

Depression symptoms among University Students in the Khulna region of Bangladesh

PLOS Mental Health

Dear Dr. tohan,

Thank you for submitting your manuscript to PLOS Mental Health. After careful consideration, we feel that it has merit but does not fully meet PLOS Mental Health’s publication criteria as it currently stands. Therefore, we invite you to submit a revised version of the manuscript that addresses the points raised during the review process.

Please ensure that your decision is justified on PLOS Mental Health’s publication criteria  and not, for example, on novelty or perceived impact.

We look forward to receiving your revised manuscript.

Kind regards,

Marc Eric Santos Reyes

Academic Editor

PLOS Mental Health

Journal Requirements:

1. Please send a completed 'Competing Interests' statement, including any COIs declared by your co-authors. If you have no competing interests to declare, please state "The authors have declared that no competing interests exist". 

3. In the online submission form, you indicated that "The datasets used and/or analysed during the current study available from the corresponding author on reasonable request.". 

3. Uploaded as supplementary information.

4. Please ensure that the Title in your manuscript file and the Title provided in your online submission form are the same.

Additional Editor Comments (if provided):

Reviewers' comments:

Reviewer's Responses to Questions

**Comments to the Author**

1. Does this manuscript meet PLOS Mental Health’s publication criteria ? Is the manuscript technically sound, and do the data support the conclusions? The manuscript must describe methodologically and ethically rigorous research with conclusions that are appropriately drawn based on the data presented.

Reviewer #1: Yes

Reviewer #2: Partly

2. Has the statistical analysis been performed appropriately and rigorously?

Reviewer #1: Yes

Reviewer #2: Yes

3. Have the authors made all data underlying the findings in their manuscript fully available (please refer to the Data Availability Statement at the start of the manuscript PDF file)?

Reviewer #1: Yes

Reviewer #2: Yes

4. Is the manuscript presented in an intelligible fashion and written in standard English?

Reviewer #1: Yes

Reviewer #2: Yes

5. Review Comments to the Author

Reviewer #1: Kindly see the attached Recommendations for Revisions.

Congratulations on your paper. Here are some of the recommendations:

1. Some statements have to be edited for grammar and completion.

2. Clearly operationalize the term/construct of depression explored in this study to avoid confusion.

3. You may include in the Introduction existing policies/programs/actions being implemented in the Khulna region that aim to mitigate depression.

4. In the Methods, what are the characteristics of these institutions? Why were they purposively chosen? Why only four institutions?

5. Based on the study's objective, the frequency (stated in the Abstract on the Cover Page, p. 1, but this is different from the Abstract on p. 5) of depression among university students was explored. How was this established by PHQ-9, considering that it only measures the participants’ depressive symptoms within two weeks? Kindly clarify.

6. Briefly describe your questionnaire for the demographics/other variables mentioned in your Results.

Reviewer #2: Introduction

Page 6:

Lines 7-8, needs in-text citation

Line 14, spell out UK first, before abbreviating

Line 16, why single-out Kenya? This statement should be deepened, backed-out with other existing literature globally. Please revise/enhance the statement.

Line 18, you mention globally, and yet, you did not provide literature proof on this. Please include in-text citations on this statement.

Lines 20-27, in a unilateral fashion, you mentioned conducted research in different countries on depression. This presentation may be improved by being more inclusive of which countries you will name or include. You also mentioned ‘in various European countries’ (line 22) but did not enumerate in the following statements whichever countries you were referring to. In your reference list, there is only one author, so how can you say Various?

Lines 28-29, continuing to Page 7, lines 1-2, you talk about a number of global studies, but your references are only from two authors (Koly et al., 2020 and Akhtarul et al., 2020) who did research in Bangladesh. How is this global? Also, this is a two-sentence paragraph, please improve/revise this.

Page 7:

Lines 3-16, Please strengthen your context, argument, justification and contribution. Your paragraph, although bearing how you give importance to mental health problems, does not give a clear picture of why among university students in the Khulna region of Bangladesh.

General Comment: What is your hypothesis? What is the main objective of this cross-sectional study? Please include in your Introduction section

Methodology

General Comment for Methodology: Before discussing the setting and population, please give a brief discussion on your Research Method. Instead of just mentioning that this is a cross-sectional study, include a sub-section, Research Method, and discuss what design/method you utilized and why the choice of this?

Page 7:

Line 20, you mentioned ‘purposively selected’ but you did not include the parameters of your research. Please include your inclusion and exclusion criteria. Also, explain what is random walk technique.

Line 24, if you randomly selected, what was your recruitment/sampling technique? Please include this in the discussion

General comment for Setting and population: Please separate your discussion on your data-collection procedure from the setting and population.

Page 8:

Line 10, introduce first what is PHQ-9, and why you chose this for your research instrument. Give a background of this assessment tool, mention the validity and reliability and how you were able to access this.

Lines 22-29, to give better context on your statistical analysis, please start your presentation of this by recalling the objective of your study.

Page 10:

Line 2, give the year of authorship for Godden

Results

General Comments:

Your Results section must be presented in accordance to the objectives of this research and not according to the Statistical Tools and Treatment. Since you were not able to clearly state the overall aim of this research, the presentation of your results in not well-presented. Please revise accordingly.

Page 12, Table 1. Please improve your table presentation. Follow the correct table formatting.

Discussion

General Comment:

Your first paragraph here should be aligned with your presentation of results. The discussion section must be parallel to your Results.

Page 16:

Lines 8-9, you mentioned about a gender angle, however, you failed to closely contextualize this with cultural aspects.

Lines 11-12, this should be cross-referenced

Line 19, you presented the finding, but you did not offer an attribution or contextualization

General Comments on Discussion:

As a general guide, please evaluate your presentation, making sure that the importance and relevance of your findings are captured, contextualized and that you have deepened the interpretation of the results in relation to your research objectives and literature review.

Page 18:

Please transfer Strengths and Limitations as a sub-section of Conclusion. Also, include in your discussion, how you were able to manage the limitations of your study and how you maximized the strengths for the success of this research.

Conclusion

General Comments:

Please enrich your conclusion. This must be aligned with your results and discussion. Include implications to theory and practice as well as future directions.

6. PLOS authors have the option to publish the peer review history of their article (what does this mean? ). If published, this will include your full peer review and any attached files.

**Do you want your identity to be public for this peer review?** For information about this choice, including consent withdrawal, please see our Privacy Policy .

Reviewer #1: No

Reviewer #2: No

---

## [Decision Letter · Decision Letter 1]

5 Nov 2024

PMEN-D-24-00166R1

Depression symptoms among University Students in the Khulna region of Bangladesh

PLOS Mental Health

Dear Dr. tohan,

Thank you for submitting your manuscript to PLOS Mental Health. After careful consideration, we feel that it has merit but does not fully meet PLOS Mental Health’s publication criteria as it currently stands. Therefore, we invite you to submit a revised version of the manuscript that addresses the points raised during the review process.

We look forward to receiving your revised manuscript.

Kind regards,

Marc Eric Santos Reyes

Academic Editor

PLOS Mental Health

Journal Requirements:

Additional Editor Comments (if provided):

Reviewers' comments:

Reviewer's Responses to Questions

**Comments to the Author**

1. If the authors have adequately addressed your comments raised in a previous round of review and you feel that this manuscript is now acceptable for publication, you may indicate that here to bypass the “Comments to the Author” section, enter your conflict of interest statement in the “Confidential to Editor” section, and submit your "Accept" recommendation.

Reviewer #1: (No Response)

Reviewer #2: (No Response)

2. Does this manuscript meet PLOS Mental Health’s publication criteria ? Is the manuscript technically sound, and do the data support the conclusions? The manuscript must describe methodologically and ethically rigorous research with conclusions that are appropriately drawn based on the data presented.

Reviewer #1: Yes

Reviewer #2: Yes

3. Has the statistical analysis been performed appropriately and rigorously?

Reviewer #1: N/A

Reviewer #2: Yes

4. Have the authors made all data underlying the findings in their manuscript fully available (please refer to the Data Availability Statement at the start of the manuscript PDF file)?

Reviewer #1: Yes

Reviewer #2: Yes

5. Is the manuscript presented in an intelligible fashion and written in standard English?

Reviewer #1: Yes

Reviewer #2: Yes

6. Review Comments to the Author

Reviewer #1: General comment:

Congratulations! Previous comments or suggestions for revision were satisfactorily incorporated in the revised version.

However, there are still nitty-gritty details for statement improvement, which their language editor may take note of.

Reviewer #2: Dear Authors,

Thank you for revising your manuscript. This first revision is better written and more substantive. I only have minor revisions:

For lines 157-158, introduce your table. Place the Table number and description before the actual Table. Improve the presentation of your table showing the sample.

For lines 159-166. introduce first your figure. Place the Figure number and description before the actual Figure. Improve the presentation of your figure showing the formula.

For lines 203-204. Place the Figure number and description before the actual Figure. Improve the presentation of your figure showing the map.

For line 218. Improve your result sub-title, Change Chi-square test to something more specific, like what did this statistical treatment measure?

For line 247. Improve your result sub-title, Change Binary logistic regression to something more specific, like what did this statistical treatment measure?

For lines 270-271. This table is adjusted to Table 2 already, since Table 1 is the sample presentation

For lines 272-274. This table is adjusted to Table 3 already, since Table 2 is the Frequency and Association of Factor Categories with Depression Symptoms.

For lines 372-387. Please move Strengths and Limitations after your Conclusion

Thank you very much and all the best!

Sincerely,

Reviewer

7. PLOS authors have the option to publish the peer review history of their article (what does this mean? ). If published, this will include your full peer review and any attached files.

**Do you want your identity to be public for this peer review?** For information about this choice, including consent withdrawal, please see our Privacy Policy .

Reviewer #1: No

Reviewer #2: **Yes: ** Salvacion Laguilles Villafuerte

---

## [Decision Letter · Decision Letter 2]

15 Jan 2025

Depression symptoms among University Students in the Khulna region of Bangladesh

PMEN-D-24-00166R2

Dear Researcher tohan,

We are pleased to inform you that your manuscript 'Depression symptoms among University Students in the Khulna region of Bangladesh' has been provisionally accepted for publication in PLOS Mental Health.

Best regards,

Marc Eric Santos Reyes

Academic Editor

PLOS Mental Health

Your paper is being accepted, however, please take note of the comments of Reviewer 1 that needs to be addressed in the final copy.

Reviewer Comments (if any, and for reference):

Reviewer's Responses to Questions

**Comments to the Author**

1. If the authors have adequately addressed your comments raised in a previous round of review and you feel that this manuscript is now acceptable for publication, you may indicate that here to bypass the “Comments to the Author” section, enter your conflict of interest statement in the “Confidential to Editor” section, and submit your "Accept" recommendation.

Reviewer #1: (No Response)

Reviewer #2: All comments have been addressed

2. Does this manuscript meet PLOS Mental Health’s publication criteria ? Is the manuscript technically sound, and do the data support the conclusions? The manuscript must describe methodologically and ethically rigorous research with conclusions that are appropriately drawn based on the data presented.

Reviewer #1: Yes

Reviewer #2: Yes

3. Has the statistical analysis been performed appropriately and rigorously?

Reviewer #1: N/A

Reviewer #2: Yes

4. Have the authors made all data underlying the findings in their manuscript fully available (please refer to the Data Availability Statement at the start of the manuscript PDF file)?

Reviewer #1: Yes

Reviewer #2: (No Response)

5. Is the manuscript presented in an intelligible fashion and written in standard English?

Reviewer #1: Yes

Reviewer #2: Yes

6. Review Comments to the Author

Reviewer #1: There are a few words that still do not follow the capitalization rules, e.g., South Asia was in lower case; and capitalized several, which does not begin a sentence.

Reviewer #2: This version improved a lot compared to the original submission. The authors may still consider having the manuscript checked/edited by a language editor.

7. PLOS authors have the option to publish the peer review history of their article (what does this mean? ). If published, this will include your full peer review and any attached files.

**Do you want your identity to be public for this peer review?** For information about this choice, including consent withdrawal, please see our Privacy Policy .

Reviewer #1: No

Reviewer #2: **Yes: ** Salvacion Laguilles Villafuerte
